# Crack Evaluation of Concrete Using Mechanochromic Sensor

**DOI:** 10.3390/ma16020662

**Published:** 2023-01-10

**Authors:** Sujeong Pyeon, Hongseop Kim, Gyeongcheol Choe, Myeongkyu Lee, Junseo Jeon, Gyuyong Kim, Jeongsoo Nam

**Affiliations:** 1Department of Architectural Engineering, Chungnam National University, 99 Daehak-ro, Yuseong-gu, Daejeon 34134, Republic of Korea; 2Department of Building Research, Korea Institute of Civil Engineering and Building Technology, Ilsanseo-gu, Goyang-si 10223, Republic of Korea; 3Department of Materials Science and Engineering, Yonsei University, 50 Yonsei-ro, Seodaemun-gu, Seoul 03722, Republic of Korea; 4Department of Geotechnical Engineering Research, Korea Institute of Civil Engineering and Building Technology, Ilsanseo-gu, Goyang-si 10223, Republic of Korea

**Keywords:** structural health monitoring system, mechanochromic sensor, strain, deformation, discoloration, concrete structure

## Abstract

In this study, the deformation of concrete materials was evaluated using a mechanochromic sensor that detects the discoloration reaction caused by deformation. This sensor was attached by applying the Loctite adhesive to both ends in the longitudinal direction. The process of applying tensile stress to the specimens was videotaped, and the deformation and discoloration were examined through image analysis. The mechanochromic sensor was not affected by the finished surface condition, and the discoloration reaction was detected for a concrete material deformation level of up to 0.01 mm. The detected level was caused by the elongation of the sensor, and the discoloration compared with the initial color was identified. In addition, the integration behavior of the mechanochromic sensor under the deterioration of concrete members in cold areas and winter environments, as well as the discoloration reaction of the sensor in a low-temperature environment, was examined. It was found that the discoloration ability of the mechanochromic sensor exposed to a low-temperature environment was restored in 2 h after the end of the freeze–thaw test, and it was judged that the deformation and discoloration levels will be properly measured when the surface temperature of the sensor is restored to a room temperature of approximately 15 °C. This appeared to be due to the room temperature recovery of the dielectric spacer of the sensor and the deformation structure of the resonance condition. The sensor was also attached when diagonal cracks occurred in the concrete beam members to evaluate the strain and discoloration rate according to the deformation and discoloration levels. Accordingly, the cracks and deformation of the concrete materials were monitored using measured values from the discoloration of the mechanochromic sensors, and the possibility of measuring the crack width was reviewed only by real-time monitoring and imaging with the naked eye.

## 1. Introduction

### 1.1. Background and Purpose of Research

The deterioration of structures, including civil engineering structures and buildings, causes significant damage worldwide every year. Maintenance and repair according to the service lives of structures are needed to reduce such damage, and various structural health monitoring (SHM) systems have been developed, including electronic endoscopes, wireless sensors, and radiofrequency identification devices [1,2,3,4,5]. SHM is a new technology in the fields of civil, mechanical, and aeronautical engineering for detecting structural damage [6]. Sophisticated installation and operation are essential for SHM systems that are installed to detect damage and cracking, and the operation is expensive and time-consuming and requires considerable manpower. Typical monitoring systems use cables to transmit data records to data pools, which has a high system cost and requires labor-intensive installation. For example, the cost can be USD 5000 per sensor channel for installing 10–15 sensors in a small system [7]. As the size of the structure increases, more sensors are required, and sensors with better robustness to the external environment are more expensive [8,9,10]. For large structures or structures to which access is impossible or difficult, sensor installation and monitoring are difficult [11]. A dense and high-precision sensor network is required for accurate monitoring of a large structure, which increases the construction cost [12]. Additionally, considerable labor is needed to manage the network, and the long construction time may make it difficult to identify the degree of damage to the structure in a timely manner [13].

Additional equipment in addition to sensors is required for most monitoring systems, and the latest sensor networking systems are in various stages of development [13]. Kang utilized smart finishing materials known as piezoelectric paint sensors, which combined polymers with piezoelectric properties and confirmed their deformation response under electric fields [14]. Such sensors, which are economical and easy to implement, significantly reduced the amount of external power used compared with conventional sensors [15]. Alternatively, the structure can be used as a sensing solution; conductive materials can be used to create a uniform conductive network inside concrete at a reasonable cost and continuously monitor damage by measuring the resistivity [16,17,18,19,20,21,22,23,24]. The materials mainly used include electrically conductive materials, such as metal powder, carbon nanofibers, and graphene [25]. A microelectromechanical system (MEMS) is an integrated system that serves as an interface between electrical and mechanical components at the microscale. Combined with many small devices (e.g., integrated circuit chips, transceivers, accelerometers, and batteries) based on microfabrication technology, the MEMS detects and processes data at the macroscale [26]. The software installed to operate the MEMS data collection system must be regularly managed, including the memory constraints of the sensor nodes, for configuration of the communication between the sensor nodes, node synchronization, measurement of the physical parameters required for SHM, sensed data processing, health and performance monitoring for each sensor, and other units of the wireless sensor network (WSN) [27,28]. As the measured data, including the damage, cracks, and deformation of structures, must be further analyzed using such software, it is necessary to simplify and optimize the maintenance monitoring process. In addition, the required electrical energy increases for large structures that require power.

Lee et al. developed a mechanochromic sensor that exhibits a discoloration reaction under the application of physical force, which has been used to measure the strain of homogeneous materials, such as steel, in the fields of machinery and aviation [5,29]. It has also been used as a hydrogel whose elasticity is reflected by its mechanical discoloration in various applications, such as modeling the curvature of body joints and fabricating elastic displays [29,30,31,32]. The mechanochromic sensor needs no power or power source for operation. In the construction field, the current SHM system monitors problems with deterioration in structures that are difficult to access or sense through image processing techniques [33]. However, structural damage assessment through image processing is applicable only to some cracks because it depends on phenomena that occur on the crack surface, such as discoloration. Therefore, the use of mechanochromic sensors in the construction field is expected to contribute to the detection of damage to structural materials. Despite this, there have been few studies in which such sensors were used for detecting damage to construction materials.

Therefore, in this study, whether the discoloration reaction of the mechanochromic sensor is active under the deformation of concrete materials was investigated to examine the applicability of the sensor in the construction field. The deformation due to cracks in concrete and structural damage were evaluated using the color change of the mechanochromic sensor, and the tensile deformation of specimens was videotaped for image analysis. In addition, the deformation and discoloration of the sensor attached to cement composites were analyzed through a freeze–thaw test, in which a freezing environment was used, to confirm its applicability to buildings in cold areas. Finally, the mechanochromic sensor was attached to the crack area during a crack induction experiment on a large concrete beam member to verify the accuracy of the concrete crack width measurement method. The crack width was measured using the color change, and the change in the sensor was determined through a comparison with the actual crack width.

### 1.2. Research Significance

In this study, mechanochromic sensors, which were not considered in the existing construction field, were intended to be used in the crack and deformation detection field of concrete materials. Unlike conventional sensors that require power, mechanochromic sensors are non-powered discoloration sensors and can be used as highly economical sensors. In addition, it is possible to detect and monitor cracks by grasping the deformation and discoloration state of the sensor only by taking images. Through this study, it is expected to expand the field of technology based on user-centered diagnostic systems.

## 2. Literature Review

The results of a literature search on existing structural integrity systems and smart sensing technologies are listed below.

Downey et al. introduced an automatic damage detection strategy through electrical conduction and power measurement using self-sensing carbon-mixed cement materials and reviewed that the proposed Monte Carlo method could detect and locate the most noticeable damage in the structure [34]. Chakraborty et al. confirmed that crack detection is possible with a 100% probability through the quality and sensitivity test of the sensor by the buried ultrasonic sensor [35]. Ahmadi et al. reviewed a sensor that could significantly reduce field costs by reducing data because the time–frequency matrix recorded via acceleration sensors reviews a new damage index and records its location only at points where it is easy to use and has a reference response [36]. Jan et al. installed detection patches to detect changes in measured strain and detect quantitative cracks and the deformation of post-tension concrete beams [37]. Tabatabaeian et al. summarized the design principles and application proposals of mechanochromism based on smart hybrid composite sensors [38]. Ahmadi et al. proposed a novel methodology based on a conical kernel distribution to validate a new damage index for the damage detection of truss bridges [39]. MSzeląg et al. reviewed the development of shape quantification technology for crack pattern detection and development and contributed to the development of non-destructive test methods [40]. Zhang et al. reviewed the effects of crack damage, repair depth, and repair materials on concrete using piezoelectric smart aggregates and monitored the wavelet packet-based crack repair process in real time to quantitatively characterize the signal energy [41]. Kocherla et al. detected cracks smaller than 10 μm under load by measuring the stress wave path generated based on the discontinuity between materials introduced by concrete cracks [42]. In a study by Berrocal et al., crack formation was detected through spatial resolution deformation program analysis, and digital image correlation programs could be measured and tracked over time, and errors were found to be less than ±3 cm and ±20 μm [43].

## 3. Experimental Design

### 3.1. Specimen Preparation

The specimens prepared to estimate the discoloration reactivity of the attached mechanochromic sensor according to the tensile deformation of concrete were fiber-reinforced cement composites at the level of Fck 40 MPa. Figure 1a shows the process of specimen preparation. The fiber-reinforced cement composites were used to prevent local fracture after cracking in the specimens and observe the discoloration reaction caused by cracking. The Portland cement used was CEM Type 1 (SsangYong C&E) of ASTM C 150-85. The standard fine aggregate in accordance with ASTM C33/C33M-18 and fly ash of the ASTM C618 class C type were used. Table 1 presents the physical and chemical properties of the binder used. The fibers used were 13 mm-long straight steel fibers with a circular cross section and an aspect ratio of 60. The steel fiber used had a density of 7.85 g/cm^3^ and a tensile strength of 2700 MPa. Table 2 presents details regarding the prepared specimens with the mix proportion that caused 1.0 mm cracks through the ductile behavior of cement composites [34].

The specimens were prepared as shown in Figure 1b–d, and they were subjected to the freeze–thaw test at 14 d of age in accordance with ASTM C666/C666M-15 to evaluate the applicability of the mechanochromic sensor in cold areas. Before the test, periodic irregularities were formed in the dog bone specimens to induce stress concentration in the tensile test.

The mechanochromic sensor used in this study is insensitive to the angle, as it has a Fabry–Perot resonator structure composed of flat thin films. Details are presented in Figure 2a. The sensor has been generally used to measure the strain of homogeneous materials, such as steel, and express mechanical discoloration as elasticity in various applications, such as modeling the curvature of body joints and fabricating elastic displays [5,29].

The mechanochromic sensor attached to identify the tensile deformation of concrete can detect microcracks, and exhibits continuous color change depending on the deformation of the attachment surface and no viewing-angle dependence [29]. It consists of a thin top metal layer and a thick bottom metal layer and generates interference of a specific wavelength owing to the phase difference of the reflected light. In this case, the reflection spectrum of the sensor surface changes with the change in color if the resonance condition changes due to the change in the dielectric spacer (styrene–butadiene–styrene layer; SBS layer) of the sensor [4,29]. The deformation of the specimen with the mechanochromic sensor can be evaluated through the color change of the sensor. According to the literature, the sensor can be attached to an aluminum specimen to visualize and quantitatively measure the surface deformation of the specimen. It is also possible to map the deformation of the surface field to which the sensor is attached through color extraction at a high resolution. This can explain the gradual color change from the initial color value when stress, such as tension, is applied to the mechanochromic sensor and structural deformation occurs.

### 3.2. Experimental Method

The overall deformation and discoloration environment of the concrete specimens with the mechanochromic sensor is presented in Figure 2. For the attachment of the sensor, an area of 10 × 7 mm^2^ at both ends of the sensor was attached to the concrete surface using Henkel’s Loctite SF 770 primer and 406 adhesive. An interval of approximately 30 s was provided after applying the adhesive before the attachment. The specimens with the sensor were installed in a direct tensile tester. After examining the tensile deformation of the tensile specimens and the discoloration reaction results of the sensor, an experiment on the deformation of concrete beam members was performed to confirm the usefulness of the deformation and discoloration of the mechanochromic sensor during the development of cracks in concrete beam members.

The freeze–thaw test was conducted in accordance with ASTM C666/C666M-15 after attaching the mechanochromic sensor to concrete specimens to evaluate its applicability in cold areas and identify the discoloration recovery time of the sensor exposed to a low-temperature environment. In the test, one cycle was performed at temperatures ranging from −18 to 4 °C, and the temperature change process was repeated for 300 cycles. During the cycle, a direct tensile test was conducted every 30 min. The test execution time and the specimen ID by measurement time are presented in Table 3. In addition, the surface temperature at the center of the attached mechanochromic sensor with respect to the measurement time was measured using a non-contact infrared thermometer, and the results are presented in Table 3.

Upon the completion of the freeze–thaw test, the tensile strain of the mechanochromic sensor and concrete, as well as the discoloration of the sensor, was measured. To evaluate the time series of deformation, the deformation of the specimens was induced using a 250-kN static direct tensile tester with the stress loading speed set as 3 mm/min. A white chamber was installed to identify the discoloration of the sensor surface according to the increase in deformation. This was to prevent visual errors by allowing a certain amount of light to fall on the sensor surface. The tensile deformations of the specimen and sensor were measured at the same position from the sensor surface. A video extensometer (Mercury, HIDAKE Technologies, Brno, Czech Republic) was used to examine the generated tensile deformation, and a quantitative measurement method satisfying the accuracy suggested in ASTM E82-10a was used. Two or more points marked with ‘+’ on the specimen surface were selected to evaluate the tensile strain of the specimen through imaging, and the tensile deformation was evaluated through image mapping. Regarding the distance between the points marked with ‘+’, two or more distances, as shown in Figure 3, were selected and entered into the software to increase the accuracy of image analysis. The measured distances for each specimen are presented in Table 4.

As shown in Figure 4, the color that appeared during the discoloration process of the mechanochromic sensor was extracted as hue, saturation, and value (HSV) to examine the overall color change. Python Ver. 3.10.8 (Python Software Foundation) was used for the color extraction. The HSV color space is a color model similar to the human visual system; thus, image processing can be achieved in a manner similar to human visual processing. In the case of the mechanochromic sensor, it was confirmed that the values of the H, S, and V components changed as the tensile strain of the concrete specimen increased. In the repeated experiments, the H value and tensile deformation exhibited a linear relationship. H expresses hue, and the linear relationship was derived because it is not affected by the expression of color, which is perceived differently depending on the brightness and saturation.

According to these results, the tensile strains of the specimen and the attached mechanochromic sensor, as well as the discoloration reaction rate of the sensor, was calculated at each measurement time after the freeze–thaw test, and correlation analysis was performed using the Statistical Package for Social Sciences version 26 (IBM SPSS Statistics 26, SPSS Inc., Chicago, IL, USA).

### 3.3. Suggestion of Discoloration Index Setting of Mechanochromic Sensor and Crack Width Derivation Formula for Estimating Deformation of Concrete

When the H (hue) component was extracted by converting the color of the part of the mechanochromic sensor where deformation and discoloration occurred into the HSV color space, its linear relationship with the concrete specimen was confirmed. However, it is difficult to specify the deformation and discoloration range of the sensor because it is not easy to capture the images of only the deformation and discoloration part of the sensor attached to the structure, which may affect the quantitative derivation of the crack width. To extract the color range of only the sensor area from the captured images of the sensor, the color analysis range was determined in the captured images, and parts other than the sensor were removed. Among the HSV components of the mechanochromic sensor surface that appear during image capture, all the H values were within the range of 20–150, and the S (saturation) and V (value) exceeded 50. Therefore, the images were divided into the 100 × 300 pixel level, and the HSV component values extracted from the pixels of the deformed area were returned according to the range, as shown in Table 5, to remove the background and show only the hue component of the sensor, as indicated by Figure 3a.

As shown in Figure 4, the discoloration reaction did not occur in all areas of the mechanochromic sensor in the captured images. In particular, no deformation or discoloration reaction occurred in the adhesive application areas. As the sensor surface with no deformation maintained the initial color, the H value of this part was the same as the initial value, which was the lowest in the sensor area. Therefore, when the deformation level of the sensor was the highest, the H value of the sensor surface color was identical to the result of the largest deformation. A formula was derived to evaluate the deformation according to the discoloration of the sensor. After defining the lowest H value in the captured image of the sensor as *H_min_* and the highest H value as *H_max_*, the ratio of the two values, i.e., *dH*, is obtained as follows:(1)$$dH=\frac{H_{max}}{H_{min}}.$$

Equation (2) gives the relationship between the calculated *dH* value and the strain. This equation can be used to quantify the discoloration and deformation characteristics of the mechanochromic sensor:(2)Strain=3.0523×dH−4.4955

In this study, the H value and deformation were calculated using Equations (1) and (2). According to the calculations, a discoloration index and crack-width derivation formula were proposed.

## 4. Results and Analysis

### 4.1. Results for Deformation and Discoloration Reaction of Mechanochromic Sensor with Concrete Specimen Exposed to Freeze–Thaw Environment

#### 4.1.1. Non-F–T

Figure 5 shows the deformation and discoloration results of the mechanochromic sensor for the concrete specimen not subjected to freezing and thawing. The initial color value of the sensor surface was green in the recorded images, and the discoloration reaction caused by deformation occurred during the extraction of the H value. Discoloration was confirmed by the transition of the value from green to purple. According to the divided images, the changes in the A1 area, A2 area, and H value for each frame were plotted on graphs, and the correlation between each area and the H value was analyzed. The A1 area and H value exhibited an R^2^ value of 0.9333, and the R^2^ value between the A2 area and H value was the same. The H value had a linear relationship with the deformation of the A1 and A2 areas, and the R^2^ value was large.

#### 4.1.2. Specimens 0.5 h after End of F–T Test

Figure 6 shows the results of analyzing the deformation and discoloration behavior of the mechanochromic sensor attached to the concrete specimen 0.5 h after the end of freezing and thawing. Frost was formed on the surface of the sensor even after the freezing and thawing, and image analysis was conducted after contaminants and moisture were removed from the surface during image capture. In the H-value extraction process, yellow represented approximately a third of the sensor surface as the initial color, and the input of the initial color value varied owing to surface frost damage. According to the divided images, the changes in the A1 area, A2 area, and H value for each frame are shown on the graphs on the right side. A color analysis of the entire area of the sensor surface indicated that no H value was extracted when the yellow section was deformed. The H value, however, was derived according to the discoloration reaction of the left deformed section that was not yellow. Accordingly, R^2^ was calculated as 0.0636 and 0.0750. As R^2^ may decrease slightly when sections with no discoloration reaction in the middle of the deformation process are considered, a process to supplement or correct this error will be required.

#### 4.1.3. Specimens 1.0 h after End of F–T

Figure 7 shows the results of analyzing the deformation and discoloration behavior of the mechanochromic sensor according to the tensile test results for the concrete specimen 1.0 h after the end of freezing and thawing. The sensor was subjected to the deformation test and image capture after the removal of foreign substances from its surface upon the completion of the freeze–thaw test. No H value was detected in most of the sensor area. This appeared to be due to the loss of discoloration reaction caused by excessive frost damage to the sensor. Accordingly, the measured *dH* level was insignificant, and the H value was constant. Therefore, R^2^ was not calculated.

#### 4.1.4. Specimens 1.5 h after End of F–T

Figure 8 shows the tensile deformation of the concrete specimen and the discoloration reaction of the mechanochromic sensor attached to it 1.5 h after the end of freezing and thawing. The surface of the sensor was relatively clean compared with other specimens after the end of freezing and thawing, but the initial color value varied, as most of the sensor area was pink or yellow in the image capture and analysis process. This appeared to be due to the frost damage of the sensor, and the color pink indicated that the sensor was affected by the discoloration of the adhesion parts caused by contact with excessive moisture. During the H-value extraction process, the initial color was orange-yellow, and most of the deformed part was changed. In addition, some of the adhesion parts of the sensor exhibited no color, indicating that the discoloration range of the sensor was narrow. *dH* exhibited reactivity throughout the deformation process, but it was smaller than the values for the specimens not exposed to freezing and thawing. R^2^ was calculated to be 0.0005 or 0.0004, indicating that the reactivity of *dH* was low compared with the A1 and A2 strains under the severe frost damage of the sensor.

#### 4.1.5. Specimens 2.0 h after End of F–T

Figure 9 shows the discoloration reactivity of the mechanochromic sensor according to the tensile deformation of the concrete specimen 2.0 h after freezing and thawing. After the end of the freeze–thaw test, the surface of the sensor was frozen and covered with frost and contaminants. The surface of the sensor was cleaned before the tensile test, and image capture was performed. Non-reproducible frost damage occurred to some parts of the sensor, and the discoloration area was slightly reduced. The area for extracting the H value was reduced, but the H value could be extracted in other areas. The strain of the A1 and A2 areas, *dH* evaluation results, and initial reaction of *dH* were confirmed. *dH* was large up to 1000 frames, but it decreased as the tensile deformation of the concrete specimen increased. This appeared to be because the discoloration reaction of the sensor became insensitive owing to frost damage. R^2^ was calculated to be 0.0135 and 0.0142.

#### 4.1.6. Specimens 2.5 h after End of F–T

Figure 10 presents the results of analyzing the relationship between the deformation of the concrete specimen and the discoloration reaction of the mechanochromic sensor 2.5 h after the end of freezing and thawing. The surface condition of the specimen and the image analysis results are shown. The sensor surface was cleaned after the end of the freeze–thaw test, and image capture/analysis was conducted. Areas close to the adhesion parts of the sensor lost discoloration reactivity. Although there were yellow areas that lost sensor reactivity on the surface of the sensor, they were small and did not significantly affect the image analysis. The discoloration reaction of the sensor could be visually examined during the H-value extraction process, as described in Section 3.2, but the *dH* value was reduced after the initial reaction. In addition, the R^2^ value was not derived because the strain of A1 and A2 was not large, and the results parallel to *dH* were calculated.

#### 4.1.7. Specimens 3.0 h after End of F–T

Figure 11 shows the deformation reaction of the mechanochromic sensor according to the tensile state of the concrete specimen 3.0 h after the end of freezing and thawing. The sensor surface was covered with less frost and contaminants compared with the other specimens after the end of freezing and thawing, and the initial color of the sensor was green. The color pink was observed in some of the sensor surface area, but it did not affect the discoloration environment, as it corresponded to the adhesion parts rather than the discoloration area. *dH* and the strain of A1 and A2 were calculated, and the results indicated that *dH* was somewhat constant. During the image capture/analysis process, green and purple colors were visually observed; thus, it was expected that the reactivity of *dH* could be evaluated as described in Section 3.2, but the actual value of *dH* was found to be different. Consequently, the discoloration reaction of the mechanochromic sensor could be visually examined 3 h after the end of freezing and thawing, but it was judged that *dH* can be determined more accurately by designing a formula that considers the temperature coefficient of the frost-damaged sensor and environmental factors, such as the amount of time at room temperature.

#### 4.1.8. Specimens 3.5 h after End of F–T

Figure 12 shows the deformation of the concrete specimen and the discoloration of the mechanochromic sensor 3.5 h after the end of freezing and thawing. As shown, the sensor surface was cleaner than those of the other specimens. Although there were yellow areas that lost sensor reactivity on the surface of the sensor, they were small and did not significantly affect the image analysis. During the H-value extraction process, the discoloration reaction of the sensor could be visually examined as described in Section 3.2, but the *dH* result could not always be determined after the initial reaction. During the H-value extraction process, the color yellow represented approximately a third of the sensor surface in the initial stage, and the input of the initial color value varied owing to surface frost damage. According to the divided images, the changes in the A1 area, A2 area, and H value for each frame are shown on the graphs on the right side. A color analysis of the entire area of the sensor surface indicated that no H value was extracted when the yellow section was deformed. The H value, however, was derived according to the discoloration reaction of the left deformed section that was not yellow. Accordingly, R^2^ was calculated to be 0.0636 and 0.075. As R^2^ may decrease slightly when sections with no discoloration reaction in the middle of the deformation process are considered, a process to supplement or correct this error will be required.

### 4.2. Validation of Crack Monitoring Technique Using Mechanochromic Sensor

#### 4.2.1. Overview of Concrete Beam Member Crack Measurement Experiment

A crack-induction experiment on large beam members was performed to validate the concrete crack width measurement method according to the aforementioned deformation and discoloration results of the mechanochromic sensor based on the tensile deformation of the concrete specimen. The experiment was conducted to identify the geometry of flexural and shear cracks in beam members according to the steel reinforcement at the concrete engineering laboratory of the Department of Civil, Architectural Engineering, and Landscape Architecture of Sungkyunkwan University. The cross-sectional area of the concrete beam was planned as H600 × B450 mm, and the clear span as D5400 mm. At the time of manufacturing the concrete beam, the slump in the unhardened state was 150 mm,150 mm, the amount of air was measured at a level of 4.5, and the amount of chloride was less than 0.30 kg/m^2^. In this study, the sensor was attached in the initial stage of cracking, and the crack width under loading was measured, as shown in Figure 13. The maximum load displacement of the concrete beam was designed to be 18.55.

#### 4.2.2. Measurement Results for Crack Width of Concrete Beam Member Obtained Using Mechanochromic Sensor

Table 6 presents a comparison between the crack width measured using a crack scale and the crack width derived from the discoloration image of the mechanochromic sensor under the progression of cracks in the beam members. As the cracks that occurred in the beam members were diagonal and did not propagate in the uniaxial tensile direction with the sensor, the sensors that exhibited a wide discoloration range were targeted. The crack width was derived using the results of the analysis focused on the discoloration image in the crack occurrence range. The crack widths of specimens A and C were similar to the values measured using the crack scale. However, for specimen B, sensor discoloration was observed in areas other than the crack occurrence area. This specimen was excluded from the analysis, but the accuracy of the crack-width value obtained by the sensor tended to decrease as the crack propagated.

The widths of the cracks in the structural members were accurately measured by analyzing the discoloration image obtained by the sensor. However, it is difficult to obtain highly reliable data owing to the manufacturing condition of the sensor, discoloration in areas other than cracks, and irregular discoloration depending on the crack geometry.

## 5. Conclusions

Conventional SHM in the construction field requires precision and considerable manpower and has a high system cost for the detection of damage and cracking. These factors have been recognized as shortcomings of monitoring and sensing technology, and various sensor networking systems have emerged in recent years. In this study, a mechanochromic sensor that exhibited discoloration under the application of mechanical and physical stress was used to evaluate structural deformation and damage. The sensor can reduce various factors required for structural health assessments, as it can perform real-time monitoring with no power. Our experimental results for the deformation and discoloration reaction of the sensor attached to concrete specimens were as follows:The relationship between the discoloration of the mechanochromic sensor and the deformation of the specimen was confirmed, and the discoloration of the sensor could be identified in real time during the progression of cracks.For the concrete specimen and mechanochromic sensor exposed to a freeze–thaw environment, the discoloration area was recognized in the sensor image 1.5 h after the end of freezing and thawing. R^2^ was also calculated from 1.5 h after the end of freezing and thawing, and it gradually increased.The mechanochromic sensor exhibited the largest R^2^ value 3.0 h after the end of freezing and thawing. When the sensor was not damaged in the freeze–thaw environment, the recovery time required for sensor discoloration measurement was estimated to be approximately 3 h. Accordingly, if images of the sensor surface condition are captured in cold areas, it is possible to identify deformation and discoloration most accurately using images after 3.0 h.The deformation and discoloration of the mechanochromic sensor with respect to the crack development in concrete beam members was analyzed, and similar tendencies were observed for the actual crack deformation and the sensor discoloration when the crack widths based on the scale and the image analysis were proportional.When the deformation of concrete materials was monitored using the mechanochromic sensor, image analysis revealed that the degree of sensor discoloration corresponded to the degree of crack development. For more accurate monitoring, it is necessary to improve the reliability of the model and the experimental environment with consideration of data collected under various conditions, such as the sensor attachment position.

This mechanochromic sensor proposes a crack deformation evaluation formula according to cracks in concrete materials, and through this, crack-width image analysis was derived. Based on these results, we plan to propose a damage index by assigning weights according to the location, shape, damage size, and range of the sensor in the future, and this research is in progress.

This study reviewed the utility of a non-powered discoloration sensor in the field of detecting cracks and deformations in concrete materials, and through this, the field of technology based on a user-centered diagnosis system will be expanded.

## Figures and Tables

**Figure 1 materials-16-00662-f001:**
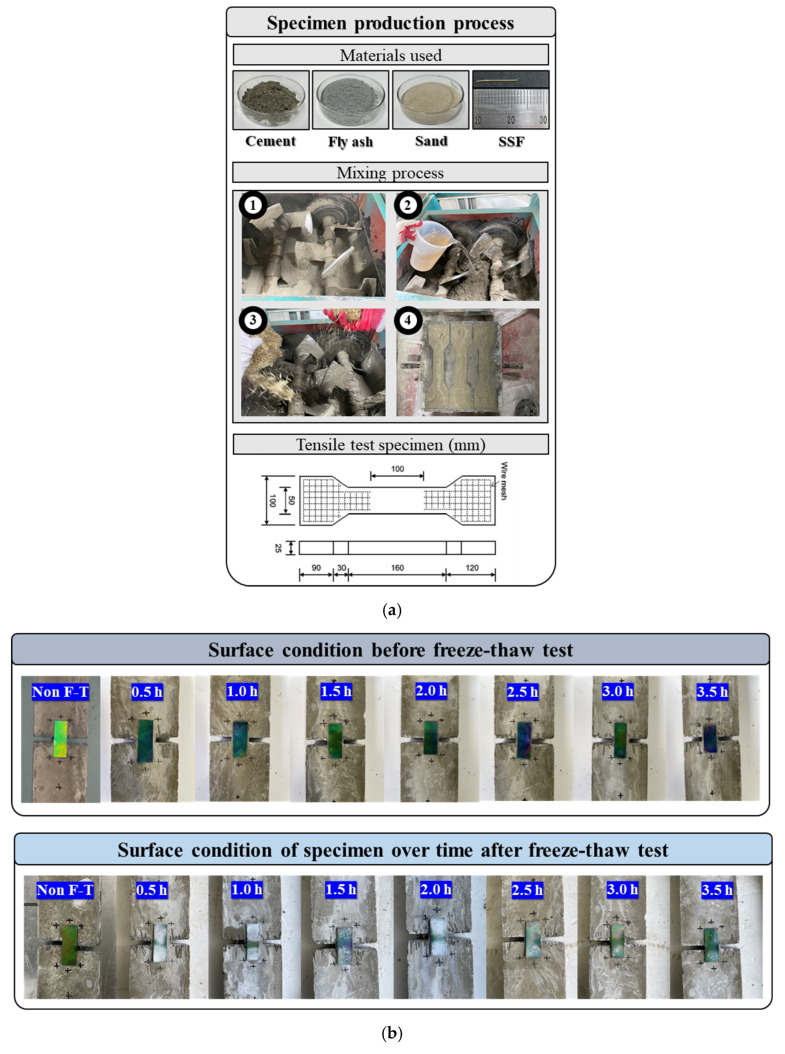
Specimen preparation and mechanochromic sensor attachment. (**a**) Details of direct tensile test specimen manufacturing process and specimen; (**b**) surface condition with mechanochromic sensor attached to cement composite; (**c**) freeze—thaw test; (**d**) temperature cycling from −18 to 4 °C for 300 cycles.

**Figure 2 materials-16-00662-f002:**
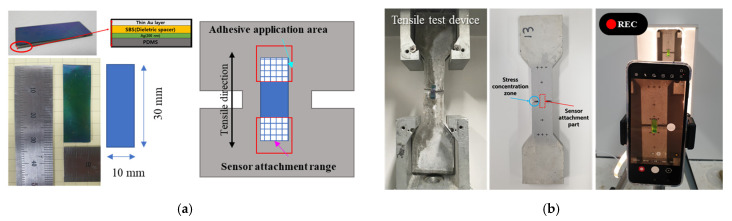
Contents of the test environment for measuring the deformation of the specimen and the discoloration response of the mechanochromic sensor. (**a**) Environment and method for attaching mechanochromic sensor; (**b**) stress concentration zone and setting for video recording.

**Figure 3 materials-16-00662-f003:**
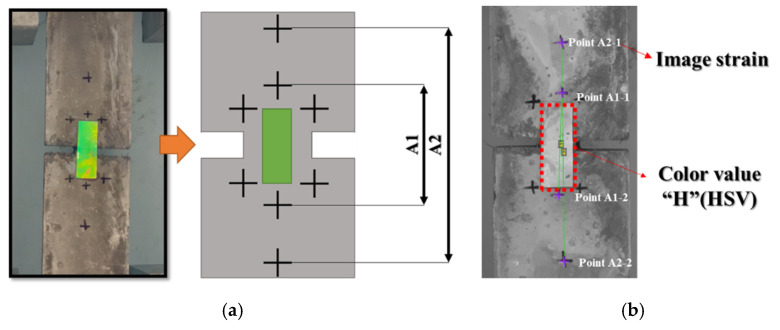
Analysis of deformation and discoloration extracted from images. (**a**) Spacing of cement composites for image elongation measurement; (**b**) deformation and discoloration analysis using images.

**Figure 4 materials-16-00662-f004:**
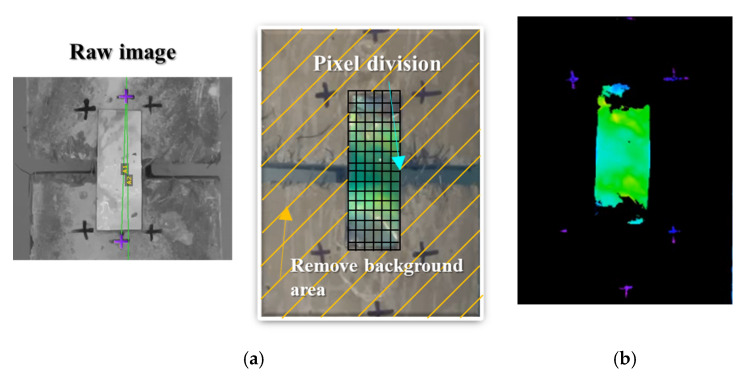
Results of pixel segmentation and background removal process for image analysis of the captured mechanochromic sensor. (**a**) Image control for improved image accuracy; (**b**) control result.

**Figure 5 materials-16-00662-f005:**
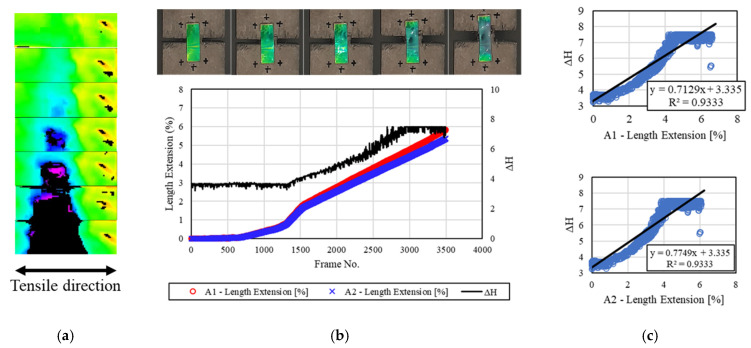
Results for the non-F–T specimen. (**a**) H-value image for discoloration of mechanochromic sensor (90° rotation relative to tensile direction); (**b**) variations in strain and *dH* of A1 and A2 according to image frame. (The discoloration process of the mechanochromic sensor attached to the specimen and the tensile strain described in Figure 3 are shown.); (**c**) correlations of *dH* with A1 and A2.

**Figure 6 materials-16-00662-f006:**
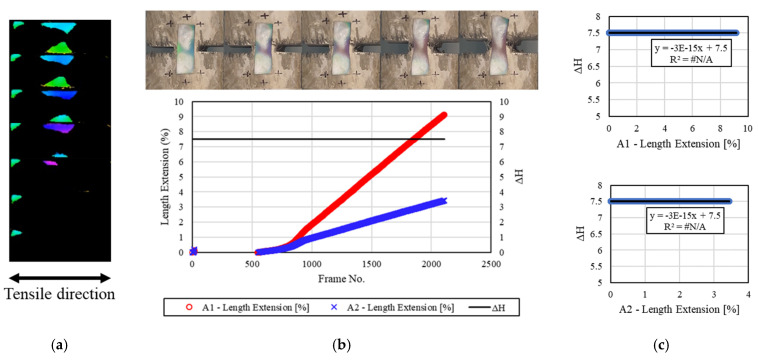
Results for the specimens after 0.5 h. (**a**) H-value image for discoloration color of mechanochromic sensor (90° rotation relative to tensile direction); (**b**) variations in strain and *dH* of A1 and A2 according to image frame. (The discoloration process of the mechanochromic sensor attached to the specimen and the tensile strain described in Figure 3 are shown.); (**c**) correlations of *dH* with A1 and A2.

**Figure 7 materials-16-00662-f007:**
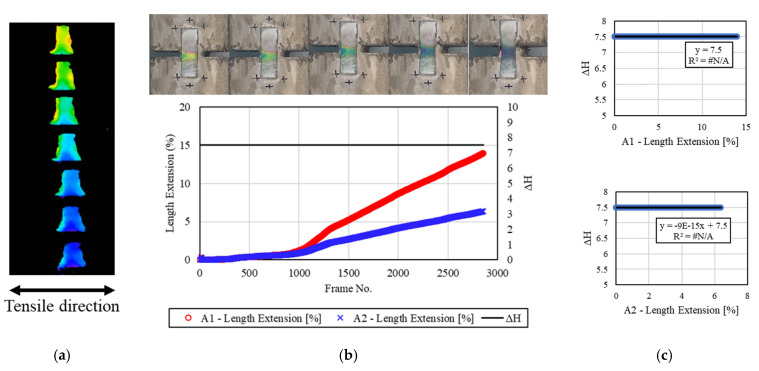
Results for the specimens after 1.0 h. (**a**) H-value image for discoloration color of mechanochromic sensor (90° rotation relative to tensile direction); (**b**) variations in strain and *dH* of A1 and A2 according to image frame. (The discoloration process of the mechanochromic sensor attached to the specimen and the tensile strain described in Figure 3 are shown.); (**c**) correlations of *dH* with A1 and A2.

**Figure 8 materials-16-00662-f008:**
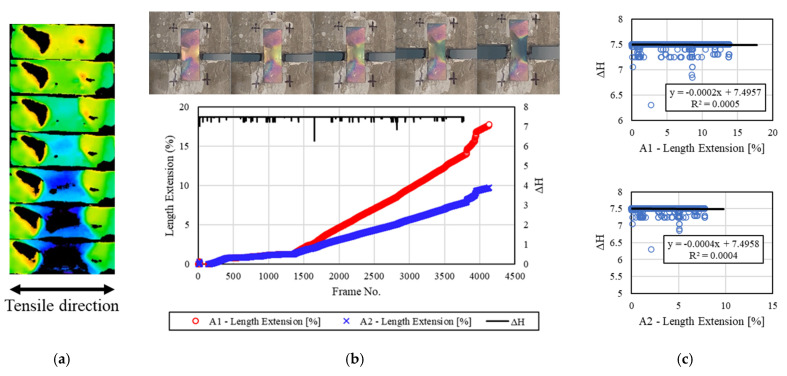
Results for the specimens after 1.5 h. (**a**) H-value image for discoloration color of mechanochromic sensor (90° rotation relative to tensile direction); (**b**) variations in strain and *dH* of A1 and A2 according to image frame. (The discoloration process of the mechanochromic sensor attached to the specimen and the tensile strain described in Figure 3 are shown.); (**c**) correlations of *dH* with A1 and A2.

**Figure 9 materials-16-00662-f009:**
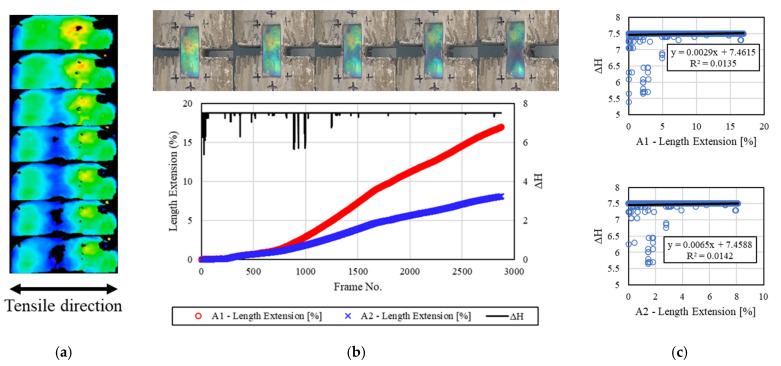
Results for the specimens after 2.0 h. (**a**) H-value image for discoloration color of mechanochromic sensor (90° rotation relative to tensile direction); (**b**) variations in strain and *dH* of A1 and A2 according to image frame. (The discoloration process of the mechanochromic sensor attached to the specimen and the tensile strain described in Figure 3 are shown.); (**c**) correlations of *dH* with A1 and A2.

**Figure 10 materials-16-00662-f010:**
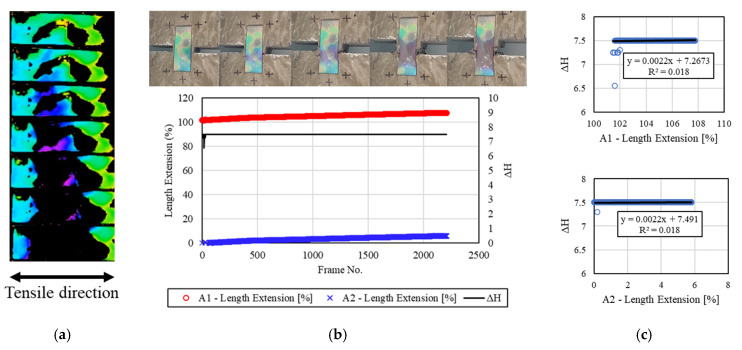
Results for the specimens after 2.5 h. (**a**) H-value image for discoloration color of mechanochromic sensor (90° rotation relative to tensile direction); (**b**) variations in strain and *dH* of A1 and A2 according to image frame. (The discoloration process of the mechanochromic sensor attached to the specimen and the tensile strain described in Figure 3 are shown.); (**c**) correlations of *dH* with A1 and A2.

**Figure 11 materials-16-00662-f011:**
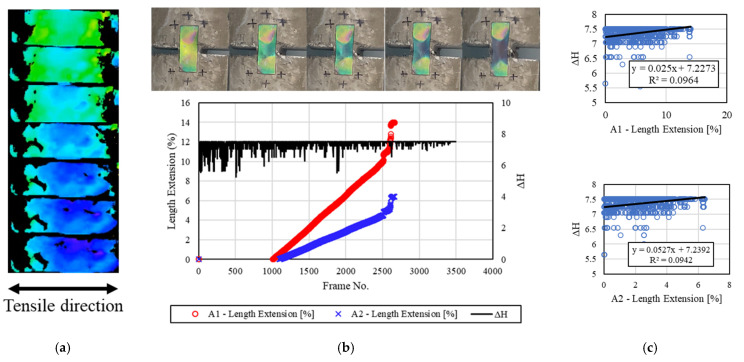
Results for the specimens after 3.0 h. (**a**) H-value image for discoloration color of mechanochromic sensor (90° rotation relative to tensile direction); (**b**) variations in strain and *dH* of A1 and A2 according to image frame. (The discoloration process of the mechanochromic sensor attached to the specimen and the tensile strain described in Figure 3 are shown.); (**c**) correlations of *dH* with A1 and A2.

**Figure 12 materials-16-00662-f012:**
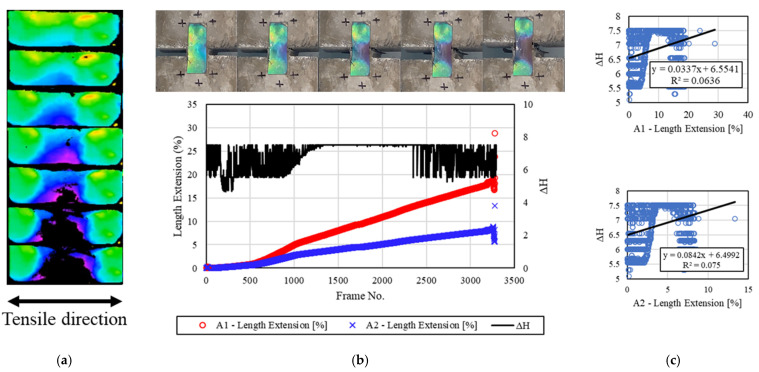
Results for the specimens after 3.5 h. (**a**) H-value image for discoloration color of mechanochromic sensor (90° rotation relative to tensile direction); (**b**) variations in strain and *dH* of A1 and A2 according to image frame. (The discoloration process of the mechanochromic sensor attached to the specimen and the tensile strain described in Figure 3 are shown.); (**c**) correlations of *dH* with A1 and A2.

**Figure 13 materials-16-00662-f013:**
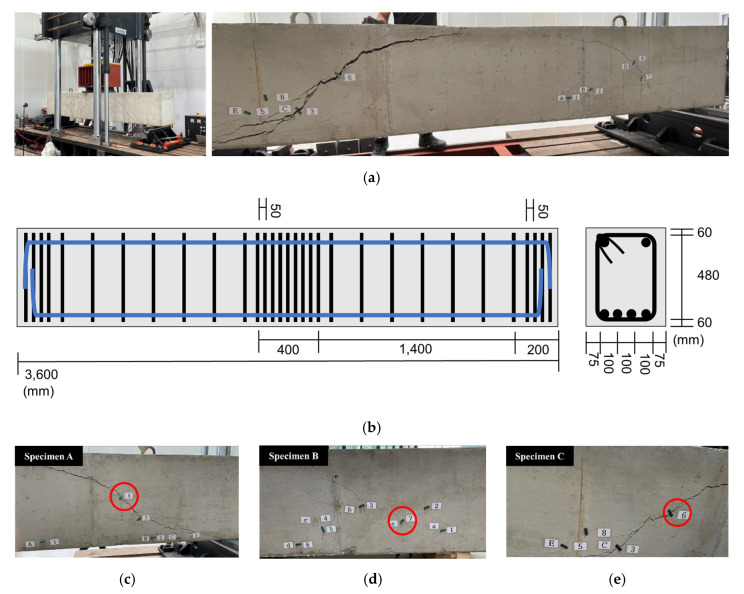
Experimental monitoring of the crack development in concrete beam members. (**a**) Concrete beam member for verification of concrete deformation of mechanochromic sensor; (**b**) design section of a concrete beam member; (**c**) specimen A; (**d**) specimen B; (**e**) specimen C.

**Table 1 materials-16-00662-t001:** Mechanical properties of the materials used.

	Type	Density (g/cm^3^)	Fineness (cm^2^/g)	Absorptance (%)
OPC	Ordinary Portland cement	3.15	3200	-
Fly ash	Class C	2.20	3000	-
Silica sand	Type 7	2.64	-	0.38

**Table 2 materials-16-00662-t002:** Mix proportion of the specimens (expressed as a ratio to the mass of cement).

F_ck_ (MPa)	W/B	Cement	Water	Fly Ash	Silica Sand	Steel Fiber
40	0.40	1.00	0.47	0.18	0.41	0.10

**Table 3 materials-16-00662-t003:** Direct tensile test execution time and sensor surface temperature for different test subjects (IDs).

Time	P (Non-F–T)	30 min	1 h	1 h 30 min	2 h	2 h 30 min	3 h	3 h 30 min
ID	No. 0	No. 1	No. 2	No. 3	No. 4	No. 5	No. 6	No. 7
Temp. (°C)	24.5	1.6	8.8	13.2	17.1	23.8	25.1	24.9

**Table 4 materials-16-00662-t004:** Distances (in mm) between the +-marked points on the surfaces of the freeze–thaw specimens used in the image analysis.

ID	No. 0	No. 1	No. 2	No. 3	No. 4	No. 5	No. 6	No. 7
Section A (mm)	40.32	41.95	42.21	41.89	39.55	40.55	42.08	40.41
Section B (mm)	101.66	102.06	103.68	103.23	103.15	101.80	102.55	101.82

**Table 5 materials-16-00662-t005:** Image background removal process for color extraction of mechanochromic sensor.

	Range	Return Value
Step 1	H < 20, H > 150, S < 50, V < 50	H, S, V = 0, 0, 0
Step 2	H > 20	H, S, V = H, 255, 255

**Table 6 materials-16-00662-t006:** Comparison of the crack-width results obtained via the image analysis and crack scale.

Sensor Position	A-4-1	A-4-2	A-4-3	A-4-4
Sensor image	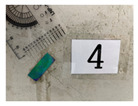	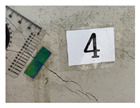	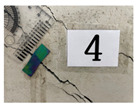	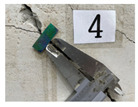
Crack width—scale (mm)	0.050	0.300	1.400	4.000
Crack width—image analysis (mm)	0.043	0.272	0.558	1.517
Sensor position	B-7-1	B-7-2	B-7-3	B-7-4
Sensor image	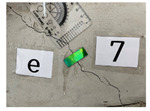	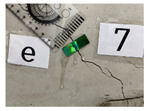	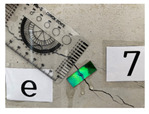	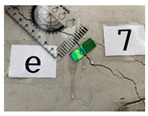
Crack width—scale (mm)	0.500	0.950	1.000	1.100
Crack width—image analysis (mm)	0.528	0.611	0.477	0.482
Sensor position	C-6-1	C-6-2	C-6-3	C6-4
Sensor image	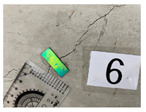	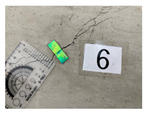	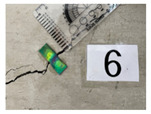	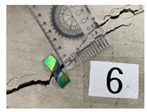
Crack width—scale (mm)	0.250	0.650	1.700	5.000
Crack width—image analysis (mm)	0.334	1.041	2.094	7.003

## Data Availability

Not applicable.

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
