# Peer review of "Crack Evaluation of Concrete Using Mechanochromic Sensor"

_materials, 2023, doi:10.3390/ma16020662_

Round 1

Reviewer 1 Report

Technically paper is good and will be useful for the readers. However, it needs to be improve to be published in Materials. It is suggested that the manuscript be rewritten according to the comments.

1-Please provide the method of extraction and additional explanations for Equation 3.

2-The explanation given for Figure 5b does not seem sufficient.

3-If reinforced concrete beams are used, its specifications should be stated. In addition, the mechanical characteristics of the materials used should be presented.

4-For specimen B of concrete beams, the color change of the sensor was observed in areas other than the crack occurrence area. Why did this happen?

5-The literature review is relatively favorable. However, it is suggested that the following papers are also reviewed and used in the manuscript.

-"Automated crack detection in conductive smart-concrete structures using a resistor mesh model." Measurement Science and Technology 29.3 (2018): 035107.

-"Early crack detection of reinforced concrete structure using embedded sensors." Sensors 19.18 (2019): 3879.

-"A novel damage identification method based on short time Fourier transform and a new efficient index." Structures. Vol. 33. Elsevier, 2021.

-"Concrete crack detection and monitoring using a capacitive dense sensor array." Sensors 19.8 (2019): 1843.

 -"Mechanical Properties, Crack Width, and Propagation of Waste Ceramic Concrete Subjected to Elevated Temperatures: A Comprehensive Study." Materials 15.7 (2022): 2371. -"A new index based on short time fourier transform for damage detection in bridge piers." Computers and Concrete 27.5 (2021): 447-455.

-"A review on self-reporting mechanochromic composites: An emerging technology for structural health monitoring." Composites Part A: Applied Science and Manufacturing (2022): 107236.

-"Health monitoring of pedestrian truss bridges using cone-shaped kernel distribution." Smart Structures and Systems 22.6 (2018): 699-709.

-"Evaluation of cracking patterns in cement composites—From basics to advances: A review." Materials 13.11 (2020): 2490.

 -"Monitoring and evaluation of the repair quality of concrete cracks using piezoelectric smart aggregates." Construction and Building Materials 317 (2022): 125775.

-"Embedded PZT sensors for monitoring formation and crack opening in concrete structures." Measurement 182 (2021): 109698.

-"Crack monitoring in reinforced concrete beams by distributed optical fiber sensors." Structure and Infrastructure Engineering 17.1 (2021): 124-139.

Author Response

The authors thank the reviewers for the time and effort that they have invested in reviewing the manuscript. We appreciate all their suggestions that have certainly improved our manuscript in multiple aspects.

All changes (including grammar review) are indicated in red text.

1-Please provide the method of extraction and additional explanations for Equation 3.

: Equation 3 is not provided in this paper. The contents of Equations 1 and 2 are written in the 342nd line, so you can refer to the corresponding contents.

2-The explanation given for Figure 5b does not seem sufficient.

: The explanation of Figure 5(b) has been added to (b). The added contents are as follows. In addition, the following information has been added from Figure 5 to Figure 12.

"The discoloration process of the mechanochromic sensor attached to the specimen and the tensile strain described in Figure 3 were shown."

3-If reinforced concrete beams are used, its specifications should be stated. In addition, the mechanical characteristics of the materials used should be presented.

: The dimension representation of concrete beams has been changed as follows. You can refer to line 613 and 616 for this.

"The cross-sectional area of the concrete beam was planned as H600×B450mm, and the clear span as D5400mm.“, “The maximum load displacement of the concrete beam was designed to be 18.55.”

4-For specimen B of concrete beams, the color change of the sensor was observed in areas other than the crack occurrence area. Why did this happen?

: I would like to explain the discoloration of the mechanochromic sensor shown in B-7-1~4 presented in Table 6.

In the currently presented photo, the discoloration other than the cracked part is not the discoloration of the mechanochromic sensor, but the discoloration caused by the interference environment of external light. Even in an environment with some light interference, the discoloration of the mechanochromic sensor in the cracked area was clearly visible, so it was possible to detect cracks.

5-The literature review is relatively favorable. However, it is suggested that the following papers are also reviewed and used in the manuscript.

: I have reviewed the references you provided and I am grateful for them. In addition, the above documents are attached to "2. Literature review".

To conclude, we look forward to hearing from all reviewers and would be happy to make further changes, if required.

Reviewer 2 Report

The paper can be published after the corrections have been made. Please send the paper again after the authors make corrections.

Author Response

The authors thank the reviewers for the time and effort that they have invested in reviewing the manuscript. We appreciate all their suggestions that have certainly improved our manuscript in multiple aspects.

All changes (including grammar review) are indicated in red text.

  1. As far as the materials used for the preparation of the samples are concerned, their properties should be mentioned as a standardized specification, especially steel fiber specifications (declared tensile strength). Please complete the mechanical properties of the fibers if they are declared.

: Based on your review, we have added the specifications of the fibers presented in the text. This is on line 158. What has been added is:

"The steel fiber used has a density of 7.85 g/cm3 and a tensile strength of 2,700 MPa."

  1. Is the specified test equipment moderate and whether the measurement deviations are in accordance with the required values?

: Measurement uncertainty representing the level of measurement quality has been calculated for this equipment, and calibration is carried out once a year to maintain precision and accuracy.

  1. To have a clearer insight into the research, it is necessary to enlarge the attached photos (Figure 1-11). Please enlarge the diagrams as well.

: There are total of 13 figures attached to the text, and the size of figures 1 to 11 has been enlarged.

  1. Please check the spelling.

: The main text was translated and proofread by a native speaker, and additional proofreading and spelling checks were carried out.

  1. Are the dimensions of the beams 600x450x540 mm in accordance with any test method - please specify.

: The dimension representation of concrete beams has been changed as follows. You can refer to line 613 for this.

"The cross-sectional area of the concrete beam was planned as H600×B450mm, and the clear span as D5400mm."

  1. Is it possible to form a damage index and relate it to the position of the sensor, the geometry of the element, the position, and the size of the damage?

: This mechanochromic sensor proposes a crack deformation evaluation formula according to cracks in concrete materials, and through this, crack width— image analysis was derived. Based on these results, a study is underway to form a damage index by assigning weights according to the position, shape, size and range of damage of the sensor in the future. This is currently in progress and will be presented in the next research paper.

To conclude, we look forward to hearing from all reviewers and would be happy to make further changes, if required.

Reviewer 3 Report

The results of “Crack evaluation of concrete using mechanochromic sensor” are of potential interest. The introduction section provides sufficient background of past literatures. In the experimental Programme section, all the testing methods are sufficiently described. In the experimental result and discussion section, the results are elaborately discussed with figures and tables. The conclusions are well presented and it is supported by the results. All the references are related to this research and also sufficient. However, the following corrections are to be carried out before the acceptance of the Manuscript.

1. Abstract: State the need of the study. Present the your research recommendation at last line.

2. Mention the novelty/research gap of your research.

3. Compare your results with existing work, if possible.

3.  What is your recommendation/future scope of your research? Present it in the conclusion section.

4. Mention your research significance/impact in the manuscript.

Author Response

The authors thank the reviewers for the time and effort that they have invested in reviewing the manuscript. We appreciate all their suggestions that have certainly improved our manuscript in multiple aspects.

All changes (including grammar review) are indicated in red text.

  1. Abstract: State the need of the study. Present the your research recommendation at last line.

: The necessity of the study was described in the abstract, which was added to the 31st line. The contents added are as follows.

Accordingly, cracks and deformation of concrete materials were monitored using measured values from discoloration of mechanochromic sensors, and the possibility of measuring crack width was reviewed only by real-time monitoring and imaging with the naked eye.

  1. Mention the novelty/research gap of your research.

: As you reviewed, we added Section “1.2 research significance” for the novelty of this study.

  1. Compare your results with existing work, if possible.

: Existing mechanochromic sensors have not been examined for the possibility of deformation and discoloration in the concrete material stage, and are mainly considered in the aviation field and single material state. This study was intended to expand the usability of the mechanochromic sensor application method in the concrete-based field.

  1. What is your recommendation/future scope of your research? Present it in the conclusion section.

: Based on your review comments, I have written and added it to the conclusion section.

The added content is as follows, presented in line 730.

“This mechanochromic sensor proposes a crack deformation evaluation formula according to cracks in concrete materials, and through this, crack width- image analysis was derived. Based on these results, we plan to propose a damage index by assigning weights according to the location, shape, damage size and range of the sensor in the future, and the research is in progress.”

  1. Mention your research significance/impact in the manuscript.

: As reviewed by the reviewers, we consider the importance and impact of this study as follows. The contents according to this have been added to the conclusion part, and you can refer to the 735 line.

“This study reviewed the utility of a non-powered discoloration sensor in the field of detecting cracks and deformations in concrete materials, and through this, the field of technology based on a user-centered diagnosis system will be expanded.”

To conclude, we look forward to hearing from all reviewers and would be happy to make further changes, if required.

Round 2

Reviewer 1 Report

Thanks to the authors.

The manuscripte can be published in the current format.